# OSCA/TMEM63 are an evolutionarily conserved family of mechanically activated ion channels

Swetha E Murthy[1], Adrienne E Dubin[1], Tess Whitwam[1], Sebastian Jojoa-Cruz[2], Stuart M Cahalan[1†], Seyed Ali Reza Mousavi[1], Andrew B Ward[2], Ardem Patapoutian[1]*

[1]Department of Neuroscience, Dorris Neuroscience Center, The Scripps Research Institute, California, United States; [2]Department of Integrative Structural and Computational Biology, The Scripps Research Institute, Howard Hughes Medical Institute, California, United States

**Abstract** Mechanically activated (MA) ion channels convert physical forces into electrical signals, and are essential for eukaryotic physiology. Despite their importance, few bona-fide MA channels have been described in plants and animals. Here, we show that various members of the OSCA and TMEM63 family of proteins from plants, flies, and mammals confer mechanosensitivity to naïve cells. We conclusively demonstrate that OSCA1.2, one of the *Arabidopsis thaliana* OSCA proteins, is an inherently mechanosensitive, pore-forming ion channel. Our results suggest that OSCA/TMEM63 proteins are the largest family of MA ion channels identified, and are conserved across eukaryotes. Our findings will enable studies to gain deep insight into molecular mechanisms of MA channel gating, and will facilitate a better understanding of mechanosensory processes in vivo across plants and animals.
DOI: https://doi.org/10.7554/eLife.41844.001

*For correspondence:
ardem@scripps.edu

Present address: †Vertex Pharmaceuticals, California, United States

Competing interests: The authors declare that no competing interests exist.

## Introduction

Mechanotransduction, the conversion of mechanical cues into biochemical signals, is crucial for many biological processes in plants and animals (*Arnadóttir and Chalfie, 2010*; *Ranade et al., 2015*). In mammals, some mechanosensory processes such as touch sensation and vascular development are mediated by the PIEZO family of mechanically activated (MA) ion channels (*Coste et al., 2010*; *Coste et al., 2012*; *Murthy et al., 2017*). For hearing, the mechanotransduction complex in hair cells includes TMC1 as the pore-forming ion channel, but also comprises of TMHS and TMIE (*Ballesteros et al., 2018*; *Pan et al., 2018*; *Qiu and Müller, 2018*). However, other MA ion channels that govern processes such as pain sensation await identification.

In plants, the impact of gravity or soil properties on root development, wind on stem growth, and turgor pressure on plant-cell size and shape are proposed to involve activation of MA ion channels (*Hamant, 2013*; *Hamant and Haswell, 2017*). Homologues of the bacterial MA channel MscS-Like (MSLs) exist in plants, and MSL8 is shown to be involved in pollen hydration in *Arabidopsis thaliana* (*Cox et al., 2015*; *Hamilton et al., 2015*). Apart from MSL8, the identity of the MA channels required for most mechanotransduction processes in plants has remained elusive. Beyond MSLs, hyperosmolarity-evoked intracellular calcium increase is shown to be dependent on the genes OSCA1.1 and OSCA1.2 in *Arabidopsis thaliana* (*Hou et al., 2014*; *Yuan et al., 2014*); however, the activation mechanism for these proteins and whether they encode a pore-forming ion channel remains unknown.

## Results

We synthesized human codon-optimized versions of OSCA1.1 (At4g04340) and OSCA1.2 (At4g22120) cDNA in pIRES2-mCherry vector, heterologously expressed them in mechanically-insensitive PIEZO1-knockout HEK293T cells (HEK-P1KO) (*Dubin et al., 2017*), and electrophysiologically characterized hyperosmolarity-activated currents. In contrast to published reports (*Hou et al., 2014*; *Yuan et al., 2014*), we find that hyperosmolarity-evoked whole-cell currents recorded from OSCA1.1- or OSCA1.2-expressing cells were only modestly larger than baseline currents (*Figure 1— figure supplement 1*).

We next explored the possibility that OSCA1.1 and OSCA1.2 are mechanosensitive, and that the modest hyperosmolarity-induced currents might be due to osmotic shock causing cell shrinking, and affecting membrane tension (*Sachs, 2010*). In cells, MA currents are commonly induced by two direct methods: 1) cell-membrane indentation with a glass probe induces macroscopic MA currents in the whole-cell patch clamp mode; 2) cell-membrane stretch induces single-channel or macroscopic MA currents when pressure is applied to a recording pipette in the cell-attached (or excised) patch clamp mode. Surprisingly, MA whole-cell currents recorded from cells transfected with OSCA1.1 or OSCA1.2 were 10- and 100-fold larger than their hyperosmolarity-activated currents, respectively (*Figure 1A,B* vs. *Figure 1—figure supplement 1*), and were comparable to those recorded from cells transfected with mouse PIEZO1, a well-characterized mechanosensitive ion channel (*Figure 1B*). Mechanosensitivity of a channel can be estimated by calculating the apparent threshold for activating MA currents that are elicited by membrane indentation. Threshold is measured as the differential of probe distance that first touches the cell and the probe distance that induces the first channel response. Therefore, it is the minimum distance of indentation required to activate the channel. OSCA1.1 and OSCA1.2 whole-cell MA currents had an apparent activation threshold of $8.6 \pm 0.9$ μm and $6.3 \pm 0.7$ μm, and inactivated (channel closure in continued presence of stimulus) with a time constant of $10.0 \pm 1.3$ ms and $10.4 \pm 1.7$ ms, respectively (*Figure 1B* and *Table 1*). Similarly, robust macroscopic stretch-activated currents were recorded from cells transfected with OSCA1.1 or OSCA1.2 but not from mock-transfected cells (*Figure 1C,D*). Stretch-activated currents from OSCA1.1 and OSCA1.2 were reversible and inactivated with a time constant of $24 \pm 3.4$ ms and $24.6 \pm 4.8$ ms, respectively (*Figure 1D*). The pressure required for half-maximal activation ($P_{50}$) of OSCA1.1 and OSCA1.2 was $-58.5 \pm 3.7$ mmHg and $-54.5 \pm 2.2$ mmHg, respectively (*Figure 1E*). These values are higher than mouse PIEZO1 which has a threshold of $-24 \pm 3.6$ mmHg (*Coste et al., 2010*; *Coste et al., 2015*) (*Figure 1E* and *Table 1*), demonstrating that at least in HEK-P1KO cells these proteins evoke high-threshold MA currents. These results suggest that OSCA1.1 and OSCA1.2 are involved in mechanotransduction.

Under physiological ionic conditions, OSCA1.1- and OSCA1.2-dependent stretch-activated currents had single-channel conductance of $184 \pm 4$ pS and $122 \pm 3$ pS, respectively (*Figure 1F,G*). We note that these values are larger than what was previously reported (*Yuan et al., 2014*). Unlike our measurements, the single-channel conductance described in *Yuan et al., 2014* was measured in the absence of a stimulus, and might not be strictly OSCA-dependent. Interestingly, stretch-activated single-channel current traces from either protein revealed a single sub-conductance state, which was half the amplitude of the full open state (*Figure 1—figure supplement 2*). Sub-conductance states are a hallmark of many ion channels and are indicative of concerted gating of multiple pores from the same channel, or of changes within a single pore (*Dani and Fox, 1991*; *Hartzell and Whitlock, 2016*; *Miller, 1982*). The presence of a single intermediate state in OSCA1.1 and OSCA1.2 could be suggestive of two cooperative subunits. Indeed, this suggestion is confirmed by the cryo-EM structures of OSCA1.1 and OSCA1.2, which reveal a pore within each subunit of a dimeric channel (*Jojoa Cruz et al., 2018*; *Zhang et al., 2018*). Characterization of the stretch-activated currents in asymmetrical NaCl solutions revealed that OSCA1.1 and OSCA1.2 evoked non-selective cation currents with some chloride permeability (OSCA1.1: $P_{Cl}/P_{Na} = 0.21 \pm 0.06$; OSCA1.2: $P_{Cl}/P_{Na} = 0.17 \pm 0.01$) (*Figure 1H*). In addition, gadolinium, a generic MA cation channel blocker, inhibited OSCA1.1- and OSCA1.2-induced stretch-activated currents (*Figure 1—figure supplement 3*). Together, these results demonstrate that OSCA1.1 and OSCA1.2 induce MA non-selective cation currents.

In *Arabidopsis thaliana*, OSCA1.1 and OSCA1.2 belong to a family of genes that include 15 members across four clades (*Hou et al., 2014*; *Yuan et al., 2014*) (*Figure 2A*). OSCA1.1 and OSCA1.2

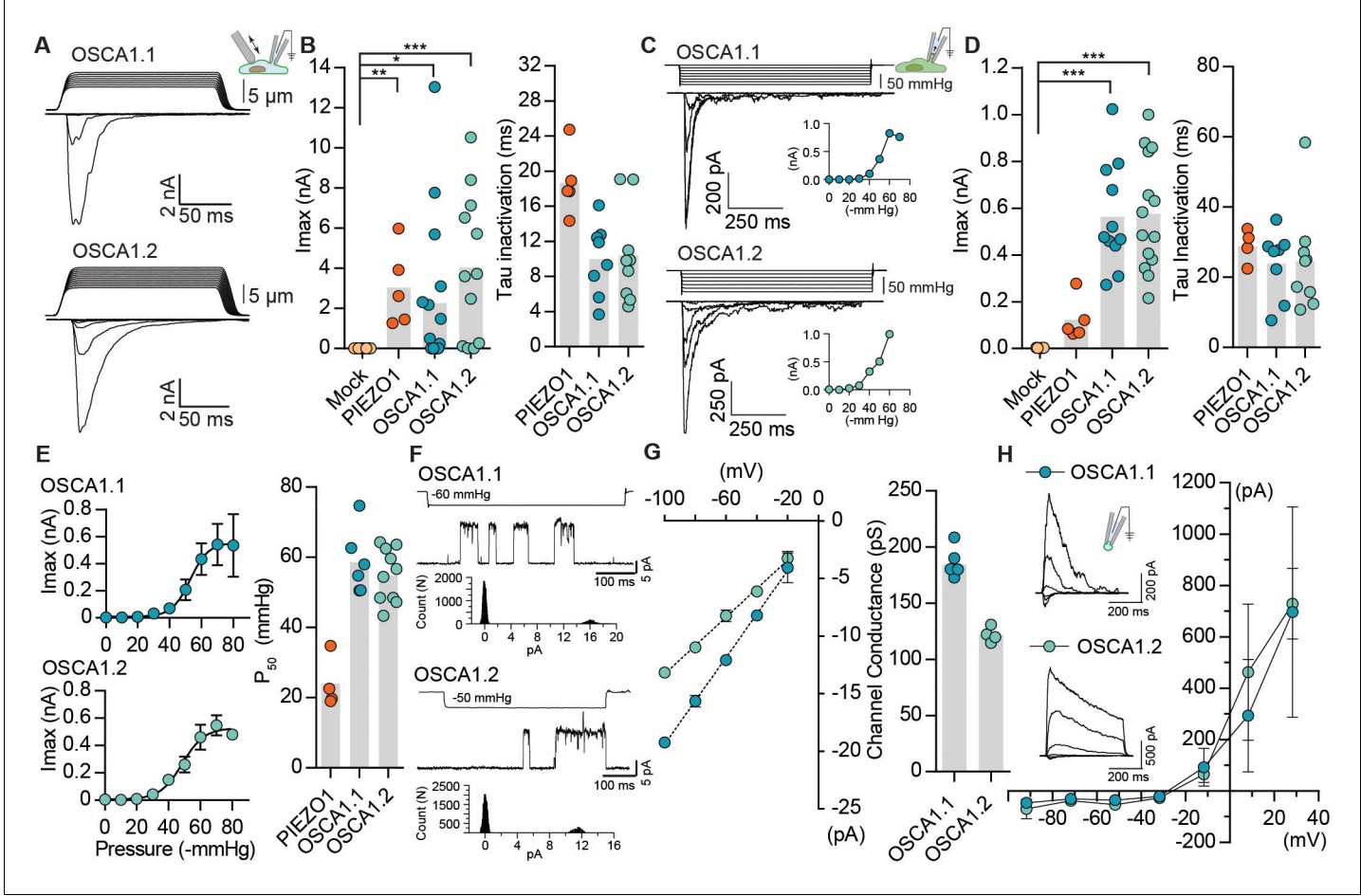

**Figure 1.** OSCA1.1 and 1.2 induce MA currents in HEK-P1KO cells. (**A**) Representative traces of MA whole-cell currents (−80 mV) from OSCA1.1- and OSCA1.2-expressing cells. The corresponding probe displacement trace is illustrated above the current trace. (**B**) Left, indentation-induced maximal currents recorded, before the patch is lost, from HEK-P1KO cells expressing mock plasmid (N = 10), MmPIEZO1 (N = 5), OSCA1.1 (N = 16, nine gave responses), or OSCA1.2 (N = 12, 10 gave responses). Right, inactivation time constant (ms) for individual cells across MmPIEZO1 (N = 5), OSCA1.1 (N = 8), and OSCA1.2 (N = 9) (*p=0.013, **p=0.005, ***p<0.0001, Dunn's multiple comparison test). (**C**) Representative traces of stretch-activated macroscopic currents (−80 mV) from OSCA1.1- and OSCA1.2-expressing cells. The corresponding pressure stimulus trace is illustrated above the current trace. Inset represents pressure-response curve for the representative cell. (**D**) Left, maximal currents recorded from HEK-P1KO cells expressing mock plasmid (N = 7), MmPIEZO1 (N = 5), OSCA1.1 (N = 11), or OSCA1.2 (N = 14). Right, inactivation time constant (ms) for individual cells across MmPIEZO1 (N = 5), OSCA1.1 (N = 8), and OSCA1.2 (N = 9) (OSCA1.1: ***p=0.0005, OSCA1.2: ***p=0.0001, Dunn's multiple comparison test). (**E**) Average pressure-response curves (fit with a Boltzman equation) for stretch-activated currents from MmPIEZO1- (N = 4), OSCA1.1- (N = 6), or OSCA1.2- (N = 10) expressing cells. Bar graph on the right represents $P_{50}$ values for individual cells across the two genes. (**F**) Representative single-channel currents (−80 mV) recorded in response to negative pipette pressure as indicated. Amplitude histogram for the trace is illustrated below. Channel openings are upward deflections. (**G**) Left, average I-V relationship of stretch-activated single-channels from OSCA1.1- and OSCA1.2-transfected cells. Right, mean channel conductance for individual cells across OSCA1.1 (N = 5) and OSCA1.2 (N = 4). (**H**) Average I-V of stretch-activated currents recorded from outside-out patches excised from cells expressing OSCA1.1 ($E_{rev}$:−24.5 ± 3.3, N = 4) or OSCA1.2 ($E_{rev}$:−25.7 ± 0.7, N = 5). Inset: representative current traces.

DOI: https://doi.org/10.7554/eLife.41844.002

The following figure supplements are available for figure 1:

**Figure supplement 1.** Hyperosmolarity-evoked currents from OSCA1.1- and OSCA1.2-transfected HEK-P1KO cells.
DOI: https://doi.org/10.7554/eLife.41844.003

**Figure supplement 2.** Subconductance states in OSCA1.1- and OSCA1.2-dependent stretch-activated single-channel traces.
DOI: https://doi.org/10.7554/eLife.41844.004

**Figure supplement 3.** Gadolinium block of OSCA1.1- and OSCA1.2-dependent MA currents.
DOI: https://doi.org/10.7554/eLife.41844.005

**Table 1.** Biophysical properties of OSCA and TMEM63 proteins.

| Gene | Whole-cell/Poke | | | Cell-attached/Stretch | | | | |
|---|---|---|---|---|---|---|---|---|
| | Imax (pA) | Inactivation tau (ms) | Threshold (μm) | Imax (pA) | Activation tau (ms) | Inactivation tau (ms) | $P_{50}$ (-mmHg) | Channel conductance (pS) |
| Mock | 4.2 ± 0.5 (10) | - | - | 2.32 ± 0.3 (17) | - | - | - | - |
| Mm PIEZO1 | 3045 ± 875 (5) | 18.7 ± 1.7 (5) | 4.0 ± 0.6 (5) | 122 ± 40 (5) | 9.0 ± 1.0 (5) | 28.8 ± 2.0 (5) | 24.0 ± 3.6 (4) | 27.3 ± 0.3 (4)* |
| OSCA1.1 | 2271 ± 918 (16) | 10.0 ± 1.3 (8) | 8.6 ± 0.9 (9) | 563 ± 68 (11) | 6.7 ± 1.0 (11) | 24.0 ± 3.0 (8) | 58.5 ± 2.4 (6) | 184.4 ± 4.4 (5) |
| OSCA1.2 | 4039 ± 1046 (12) | 10.4 ± 1.7 (9) | 6.3 ± 0.7 (10) | 576 ± 65 (14) | 5.5 ± 0.5 (14) | 24.6 ± 4.7 (9) | 54.5 ± 2.2 (10) | 121.8 ± 3.4 (4) |
| OSCA1.8 | 7.1 ± 0.7 (11) | - | - | 347 ± 56 (7) | 14.0 ± 2.4 (7) | 67.0 ± 13.0 (6) | 79.3 ± 9.9 (6) | 46.6 ± 2.8 (4) |
| OSCA2.3 | 54.34 ± 48.9 (11) | - | - | 31.3 ± 11 (7) | 18.8 ± 3.2 (6) | 110.0 ± 70.0 (6) | 59.7 ± 3.8 (7) | n.d. |
| OSCA3.1 | 9.7 ± 1.4 (7) | - | - | 306 ± 47 (12) | 5.6 ± 1.4 (8) | 18.5 ± 2.5 (11) | 44.5 ± 3.2 (5) | 24.9 ± 3.4 (4) |
| OSCA4.1 | 13.3 ± 2.0 (6) | - | - | 1.2 ± 0.05 (6) | - | - | - | - |
| Dm TMEM63 | 9.8 ± 2.0 (11) | - | - | 14.2 ± 4.5 (9) | 22.6 ± 4.3 (5) | 154 ± 34 (5) | 90 ± 2 (3) | n.d. |
| Mm TMEM63A | 2.0 ± 1.2 (13) | - | - | 18.75 ± 3.0 (23) | 126 ± 19 (8) | 323 ± 30 (9) | 60 ± 5.4 (6) | n.d. |
| Mm TMEM63B | 3.8 ± 1.8 (7) | - | - | 17.97 ± 3.9 (12) | 245 ± 38 (7) | 323 ± 30 (7) | 66 ± 7.8 (4) | n.d. |
| Mm TMEM63C | 4.5 ± 0.9 (8) | - | - | 4.73 ± 0.8 (7) | - | - | - | - |
| Hs TMEM63A | 4.2 ± 0.7 (6) | - | - | 12.4 ± 4.4 (9) | 188 ± 18 (5) | 427 ± 94 (2) | 59.7 ± 3.2 (5) | n.d. |

Note: n.d.: not determined. Ns are indicated in parenthesis. Imax value is reported for the last indentation or stretch stimulus before losing the cell. Activation is reported as 10–90% rise of stretch-activated current at saturating stimulus. Inactivation time constant for stretch-activated currents are reported in the range of −60 to −80 mmHg stimulus-pressure. All values are mean ± s.e.m. * values as indicated in **Saotome et al., 2018**

DOI: https://doi.org/10.7554/eLife.41844.006

share 85% sequence identity, while other family members within the same clade are more divergent (50 – 70%). Genes within Clade 2 have about 30% identity to Clade 1, and Clade 3 and 4 share the least homology with Clade 1 (*Figure 2A* and *Table 2*). We investigated whether mechanosensitivity might be conserved across this family. We selected one gene from each clade, and tested whether their expression could induce MA currents in a heterologous expression system (*Figure 2*). In the whole-cell patch clamp mode, mechanical indentation of cells expressing OSCA1.8, OSCA2.3, OSCA3.1, or OSCA4.1 did not elicit MA currents (*Figure 2—figure supplement 1*). Remarkably, however, distinct stretch-activated currents were recorded from cell expressing OSCA1.8, OSCA2.3, or OSCA3.1, but not OSCA4.1 (*Figure 2B,C* and *Table 1*). Although the lack of OSCA4.1-induced MA currents could suggest that this gene is functionally distinct from other members of the family, we cannot rule out that OSCA4.1 might be incorrectly folded or not trafficked to the membrane in HEK-P1KO cells. The three mechanosensitive OSCA proteins had disparate gating kinetics, with different inactivation time constants (*Figure 2C*). While the pressure of half-maximal activation was comparable among all the mechanosensitive OSCA proteins (*Figure 2D*), the most striking feature was the diversity in their single-channel conductance (*Figure 2E,F*). OSCA1.8 and OSCA3.1 channel conductance was approximately four- and six-fold smaller than OSCA1.1, respectively. Furthermore, single-channel amplitude of OSCA2.3 stretch-activated currents were in the sub-picoampere range and unresolvable, which could explain the relatively smaller maximal current responses (*Figure 2C*). These results provide evidence that OSCAs are a family of MA ion channels with unique biophysical properties. Intriguingly, the different members of the OSCA family have differential responses to the

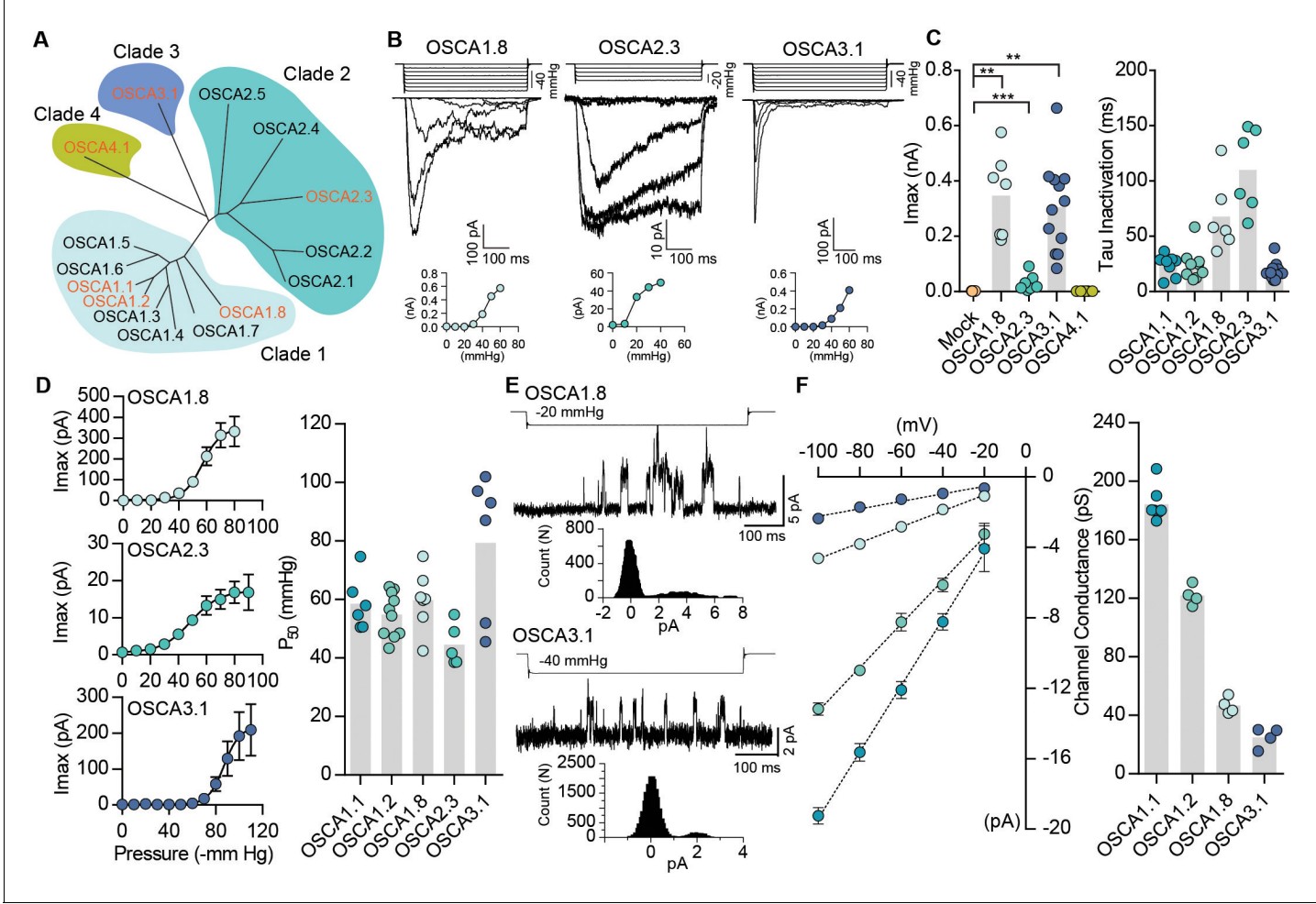

**Figure 2.** Distinct subclasses of OSCA family members induce MA currents in HEK-P1KO cells. (A) Phylogenetic tree describing sequence relationship between the 15 OSCA family members. Protein sequences were aligned using MegAlign Pro and tree was generated using DrawTree. Genes in orange were selected for characterization of mechanically-induced biophysical properties of the channel. (B) Representative traces of stretch-activated macroscopic currents recorded from cells expressing OSCA1.8, OSCA2.3, or OSCA3.1. The corresponding stimulus trace is illustrated above the current traces. Pressure-response curve for the representative cell is illustrated below the current traces. (C) Left, maximal current recorded from individual cells expressing mock plasmid (N = 7), OSCA1.8 (N = 7), OSCA2.3 (N = 7), OSCA3.1 (N = 12), or OSCA4.1 (N = 6). Right, inactivation time constant for individual cells expressing OSCA1.1 (N = 8), OSCA1.2 (N = 9), OSCA1.8 (N = 6), OSCA2.3 (N = 6), or OSCA3.1 (N = 11) (OSCA1.8: **p=0.004, OSCA3.1: **p=0.003, OSCA2.3:***p=0.0006, Dunn's multiple comparison or Mann-Whitney tests). (D) Average pressure-response curves fit with the Boltzmann equation for OSCA1.8, OSCA 2.3, and OSCA3.1. Individual $P_{50}$ values for cells expressing each protein are plotted on the right (OSCA1.1 (N = 6), OSCA1.2 (N = 10), OSCA1.8 (N = 7), OSCA2.3 (N = 5), and OSCA3.1 (N = 6)). (E) Representative stretch-activated single-channel currents recorded at −80 mV from cells expressing OSCA1.8 or OSCA3.1. Channel openings are upward deflections. The stimulus trace for the current is illustrated above. Amplitude histogram for the trace is depicted below. (F) Average single-channel I-V curves and slope conductance for the indicated protein. OSCA1.1: N = 5, OSCA1.2: N = 4, OSCA1.8: N = 4, and OSCA3.1: N = 4. OSCA1.1 and OSCA1.2 data from *Figure 1* is replotted in this figure for comparison.
DOI: https://doi.org/10.7554/eLife.41844.007

The following figure supplement is available for figure 2:

**Figure supplement 1.** Membrane indentation-induced MA currents in HEK-P1KO cells transfected with different genes from the OSCA family.
DOI: https://doi.org/10.7554/eLife.41844.008

two types of mechanical stimulation. It is noteworthy that OSCA1.1 and OSCA1.2 which were the only channels activated by membrane indentation, had a high threshold (8 μm) compared to mouse PIEZO1 (4 – 5 μm) (*Dubin et al., 2017*) (*Table 1*). Perhaps other OSCA channels have a much higher threshold, technically limiting us from recording reliable MA currents, since HEK-P1KO cells often break near 10 – 12 μm of indentation. Alternatively, mechanical indentation and membrane

**Table 2.** Percent identity in protein sequence among OSCA and TMEM63 family of genes.

| Gene | Osca 1.1 | Osca 1.2 | Osca 1.8 | Osca 2.3 | Osca 3.1 | Osca 4.1 | Dm TMEM63 | Mm TMEM63A | Mm TMEM63B | Mm TMEM63C | Hs TMEM63A |
|---|---|---|---|---|---|---|---|---|---|---|---|
| OSCA 1.1 | 100 | 84.81 | 58 | 32.76 | 29.09 | 18.82 | 18.07 | 19.19 | 19.32 | 17.55 | 20.11 |
| OSCA 1.2 | - | 100 | 58.39 | 34.47 | 28.95 | 19.42 | 18.40 | 20.09 | 19.52 | 17.75 | 20.43 |
| OSCA 1.8 | - | - | 100 | 32.24 | 31.79 | 18.00 | 18.32 | 19.26 | 19.06 | 19.15 | 19.46 |
| OSCA 2.3 | - | - | - | 100 | 23.80 | 17.30 | 17012 | 19.57 | 18.17 | 17.88 | 19.41 |
| OSCA 3.1 | - | - | - | - | 100 | 18.39 | 18.92 | 22.36 | 20.33 | 19.79 | 21.75 |
| OSCA 4.1 | - | - | - | - | - | 100 | 18.97 | 21.27 | 21.30 | 19.68 | 21.42 |
| Dm TMEM63 | - | - | - | - | - | - | 100 | 32.58 | 32.79 | 30.78 | 32.59 |
| Mm TMEM63A | - | - | - | - | - | - | - | 100 | 57.16 | 41.28 | 90.05 |
| Mm TMEM63B | - | - | - | - | - | - | - | - | 100 | 44.46 | 57.58 |
| Mm TMEM63C | - | - | - | - | - | - | - | - | - | 100 | 40.73 |
| Hs TMEM63A | - | - | - | - | - | - | - | - | - | - | 100 |

DOI: https://doi.org/10.7554/eLife.41844.009

stretch deliver biophysically-distinct forces to the membrane and the different proteins might be tuned to the unique type of force applied.

The results described so far indicate that OSCAs evoke stretch-activated currents in a heterologous expression system, meeting at least one of the criteria for MA channels. Indeed, these results by themselves only allow us to conclude that they are a component of an ion channel; it is formally possible that they require endogenously expressed components within HEK-P1KO cells to behave as a functional mechanosensitive ion channel (*Arnadóttir and Chalfie, 2010*). To address this possibility, we purified OSCA1.2 fused to a GFP tag, and reconstituted the protein in liposomes to determine whether stretch-activated currents can be recorded from excised proteoliposome patches (*Figure 3A*). Purified OSCA1.2-EGFP (117 kDa) appears as a single protein band on a denaturing gel, indicating the absence of other associated proteins (*Figure 3—figure supplement 1*). Before characterizing OSCA1.2, we first tested whether we could successfully record stretch-activated currents from liposomes reconstituted with the bacterial mechanosensitive ion channel *E. coli* MscL (*Figure 3—figure supplement 1*) (*Häse et al., 1995*). We observed MscL-like stretch-induced channel activity in seven out of eight excised patches (*Figure 3B–D*). The stretch-activated single-channel currents had a conductance of 3340 ± 180 pS (N = 6), which is in accordance with previous reports (*Kloda and Martinac, 2002*) (*Figure 3C*). Remarkably, applying negative pipette-pressure in the range of 0 to 50 mmHg to patches excised from liposomes reconstituted with OSCA1.2-EGFP protein also induced robust macroscopic currents (*Figure 3E,F*). Applying negative pressure (as high 100 to 120 mmHg) to patches excised from empty liposomes did not change the baseline current. At the single-channel level, the stretch-activated currents exhibited sub-conductance states, were voltage-dependent, and had a conductance of 346 ± 13 pS in 200 mM KCl solution, which matched the single-channel conductance measured from OSCA1.2-expressing cells (304 ± 9 pS) under the same ionic concentration (*Figure 3—figure supplement 1* and *Figure 3G,H*). These results demonstrate that OSCA1.2, like other bona-fide MA ion channels such as MscL, TREK and TRAAK, and PIEZO1, is directly gated by changes in membrane tension, and requires no additional cellular components for activation (*Anishkin et al., 2014*; *Brohawn et al., 2014*; *Häse et al., 1995*; *Syeda et al.,*

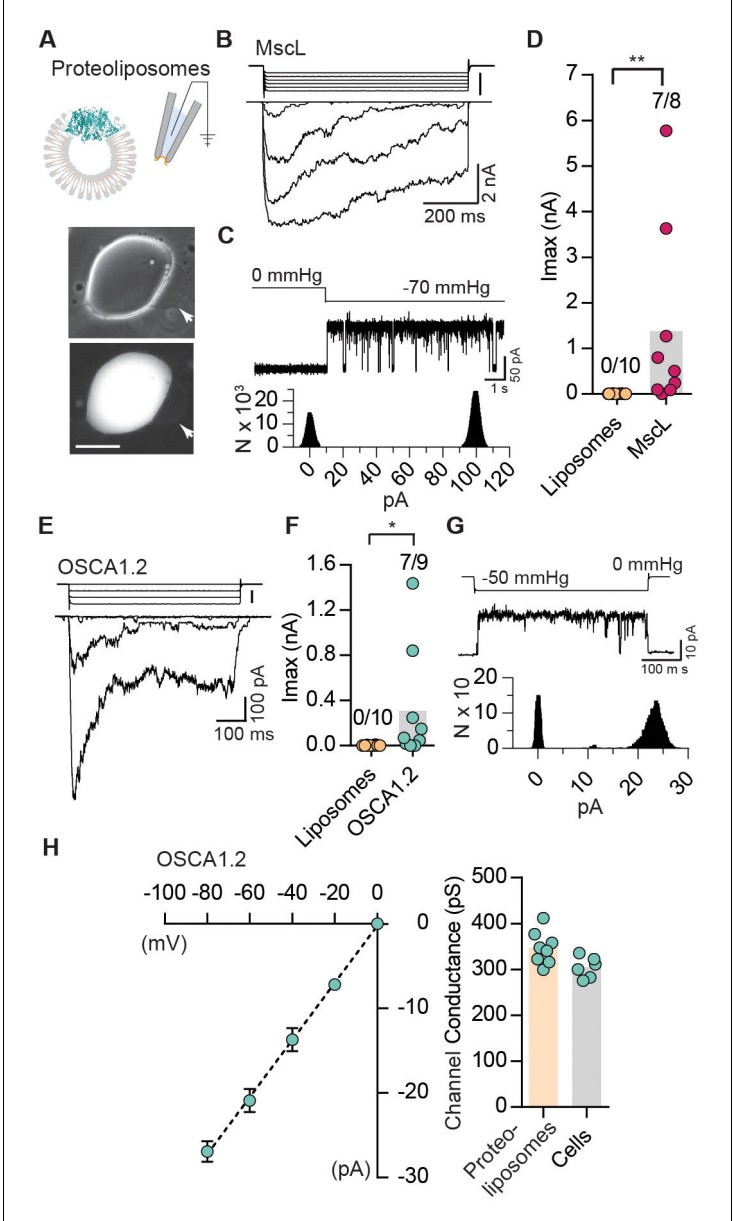

**Figure 3.** Reconstituted OSCA1.2 in liposomes form MA ion channels. (**A**) Illustration to depict that patches were pulled from liposomes reconstituted with purified MscL or OSCA1.2 proteins. Channels in patch were activated by negative pipette pressure. Brightfield (top) and GFP (bottom) images of proteoliposomes reconstitutes with OSCA1.2-EGFP protein. Patches were pulled from unilamellar vesicles (indicated by arrows). Scale bar: 50 μm. (**B**) Representative traces of macroscopic stretch-activated currents recorded from unilamellar liposomes reconstituted with EcMscL. The corresponding negative pipette pressure-stimulus is illustrated above the current traces. (**C**) Representative single-channel trace recorded in response to −70 mmHg pressure. Channel openings are upward deflections. Currents were filtered at 10 kHz. Amplitude histogram of the full-trace is depicted below. (**D**) Maximal stretch-activated currents recorded from empty liposomes or EcMscL reconstituted liposomes. Fractions indicate attempts that resulted in currents/total number of attempts (**p=0.003, Mann-Whitney test). (**E**) Representative traces of macroscopic stretch-activated currents recorded from liposomes reconstituted with OSCA1.2-EGFP protein. (**F**) Maximal stretch-activated currents recorded from empty liposomes or OSCA1.2-EGFP (*p=0.04, Mann-Whitney test). (**G**) Representative trace and amplitude histogram of stretch-activated OSCA1.2-EGFP single-channel currents recorded from proteoliposomes. Currents were filtered at 2 kHz. (**H**) Left, average I-V relationship of stretch-activated single-channels in liposomes reconstituted with OSCA1.2-EGFP. Right, single-channel conductance in 200 mM KCl solution from individual proteoliposome patches (N = 8) or cells expressing OSCA1.2 (N = 6). In (**C**) and (**G**), N represents count or number of events.

*Figure 3 continued on next page*

*Figure 3 continued*

DOI: https://doi.org/10.7554/eLife.41844.010

The following figure supplement is available for figure 3:

**Figure supplement 1.** Purified OSCA1.2 induces stretch-activated single-channel currents with subconductance states when reconstituted in liposomes.

DOI: https://doi.org/10.7554/eLife.41844.011

*2016*). This represents conclusive evidence that OSCA1.2 encodes a pore-forming subunit and is inherently mechanosensitive (*Jojoa Cruz et al., 2018*).

We next examined if orthologues of OSCAs are also mechanosensitive. Phylogenetic analysis and previous studies have identified the TMEM63 family of proteins as the closest homologues of OSCAs (*Hou et al., 2014*; *Zhao et al., 2016*; *Yuan et al., 2014*) (*Figure 4A* and *Table 2*). To determine whether mechanosensitivity was conserved across different species, we selected homologues in fruit fly, mouse, and human, and tested their ability to induce MA currents in HEK-P1KO cells. In the whole-cell patch clamp mode, mechanical stimulation of cells expressing DmTMEM63, MmTMEM63A, MmTMEM63B, MmTMEM63C, or HsTMEM63A did not elicit MA currents (*Figure 4—figure supplement 1*). However, these clones (with the exception of MmTMEM63C) induced stretch-activated currents when expressed in naïve cells (*Figure 4B,C*). The phylogenetic tree illustrates that mammalian TMEM63C is divergent from TMEM63A and TMEM63B (*Figure 4A*), which may explain its lack of mechanosensitivity. Alternatively, it is possible that this protein is incorrectly folded and not trafficked to the membrane, or that TMEM63C is not sufficient by itself to induce MA currents.

$P_{50}$ values for the TMEM63-induced stretch-activated currents were in the range of -60 to -80 mmHg, suggesting that like OSCAs, these proteins elicit high-threshold MA currents in HEK-P1KO cells (*Figure 4E*). The stretch-activated single-channel currents induced by either DmTMEM63 or the mammalian TMEM63s are in the sub-picoampere range and were unresolvable, similar to OSCA2.3 (*Figure 4B* vs. *Figure 2B*). However, compared to plant OSCAs, MA currents induced by the TMEM63 family members had unique gating properties with several fold slower activation and inactivation kinetics (*Figure 4D* and *Table 1*). MmTMEM63A stretch-activated currents are non-selective cationic, and gadolinium inhibits MmTMEM63A-induced MA currents by 75% (*Figure 4—figure supplement 2*). The stark difference in MA current properties between OSCA2.3/TMEM63s and other OSCA members is interesting. At least MmTMEM63A is expressed at the membrane (*Figure 4—figure supplement 3*), therefore trafficking issues cannot account for the relatively small stretch-activated macroscopic currents. However, one cannot exclude the possibility that in the native cellular environment, other auxiliary subunits might interact with OSCA2.3 and TMEM63 proteins to alter the channel's permeation and gating properties. Future studies will test this hypothesis. Nonetheless, these results demonstrate that orthologues of OSCAs also induce stretch-activated currents when overexpressed in naïve cells, and that mechanosensitivity is conserved across the various members in the OSCA/TMEM63 family.

## Discussion

We provide evidence that members of OSCA family are bona-fide, pore-forming mechanosensitive ion channels: (1) transfection of various members of the family from plants, flies, and mammals give rise to robust MA currents. Importantly, single-channel conductance of individual OSCAs are quite distinct, arguing that the OSCAs contribute to pore properties of these currents. (2) We directly show in vitro (proteoliposomes) that OSCA1.2 is an inherently mechanosensitive ion channel in the absence of other proteins. (3) The accompanying paper describes the high-resolution structure of OSCA1.2 (*Jojoa Cruz et al., 2018*), and demonstrates that this protein has similar architecture to the TMEM16 family of ion channels (*Whitlock and Hartzell, 2017*). Furthermore, structure-guided mutagenesis verifies that a residue within the putative pore-forming region contributes to single-channel conductance (*Jojoa Cruz et al., 2018*). Therefore, with 15 different members present just in *Arabidopsis thaliana* (5/6 members we tested were mechanosensitive), OSCA/TMEM63 proteins potentially represent the largest family of mechanosensitive ion channels known to date. Although other eukaryotic ion channels have been previously described, none are conserved from plants to

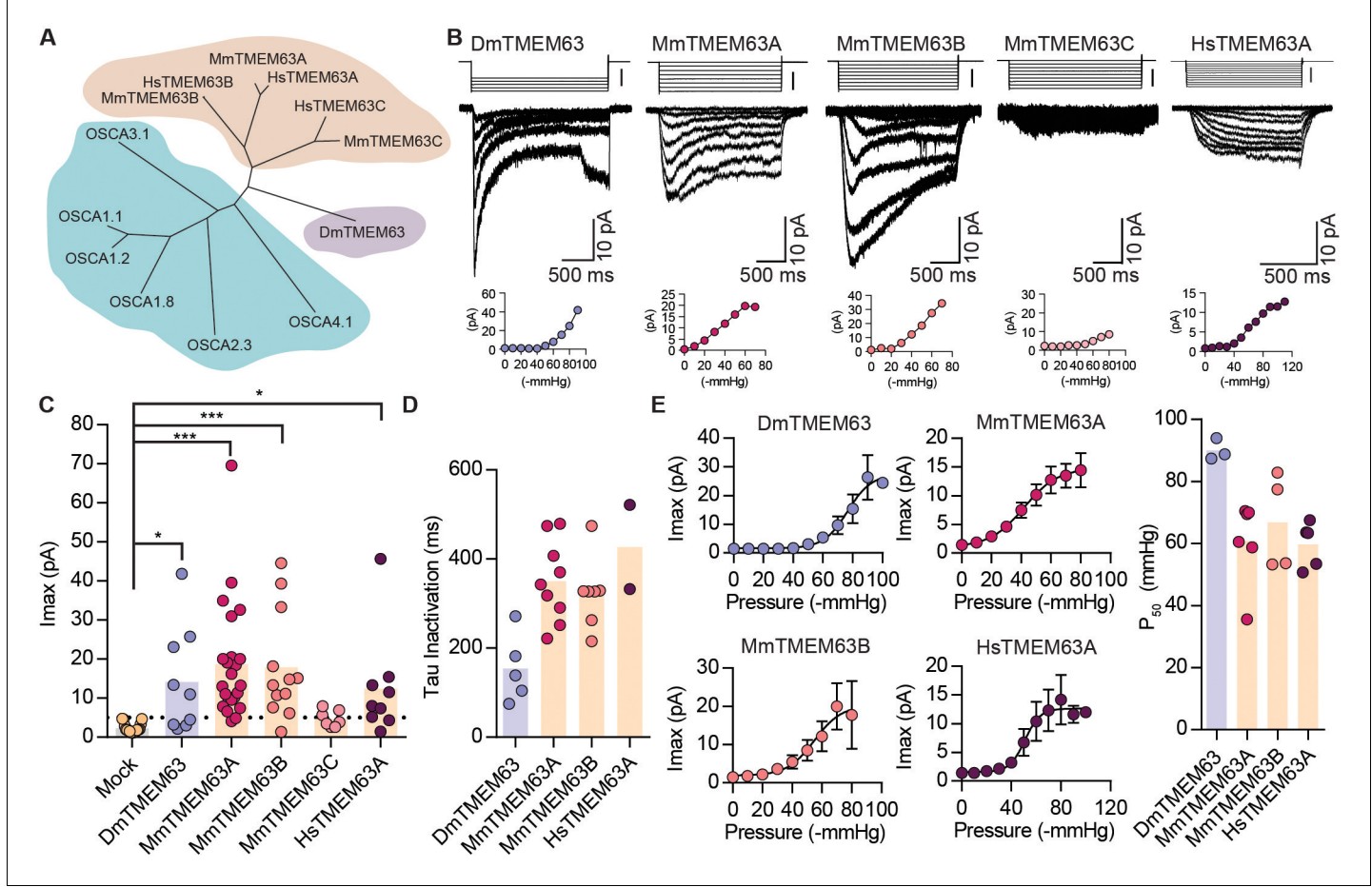

**Figure 4.** OSCA orthologues in flies and in mammals induce MA currents in HEK-P1KO cells. (A) Phylogenetic tree illustrating the relationship between OSCA proteins across *Arabidopsis thaliana* (teal clade), *Drosophila melanogaster* (purple clade), and *Mus musculus* and *Homo sapiens* (orange clade). Sequences were aligned in MegAlign Pro and tree was generated in DrawTree. (B) Representative stretch-activated currents induced by negative pipette pressure from cells transfected with the indicated gene. Corresponding pressure stimulus is illustrated above the current trace. Vertical scale bar: −50 mmHg. Pressure-response curve for the representative cell is illustrated below the trace. (C) Maximal stretch-activated currents recorded from cells expressing mock plasmid (N = 17) or the indicated TMEM63 homologue (DmTMEM63 (N = 7), MmTMEM63A (N = 23), MmTMEM63B (N = 12), MmTMEM63C (N = 7), and HsTMEM63A (N = 6). DmTMEM63: *p=0.023, HsTMEM63A: *p=0.015, ***p<0.0001, Dunn's multiple comparison test relative to mock plasmid). (D) Inactivation time constant (ms) across individual cells for each transfected gene. Owing to the non-inactivating nature of HsTMEM63A currents, only 2 out of 9 cells could be reliably fit with an exponential curve (DmTMEM63 (N = 5), MmTMEM63A (N = 9), MmTMEM63B (N = 7), and HsTMEM63A (N = 2)). (E) Left, average pressure-response curves fit with the Boltzmann equation for the indicated transfected gene. Right, $P_{50}$ values for individual cells across the four different TMEM63 homologues (DmTMEM63 (N = 3), MmTMEM63A (N = 6), MmTMEM63B (N = 4), and HsTMEM63A (N = 5)).

DOI: https://doi.org/10.7554/eLife.41844.012

The following figure supplements are available for figure 4:

**Figure supplement 1.** Membrane indentation-induced MA currents in HEK-P1KO cells transfected with TMEM63 homologues.
DOI: https://doi.org/10.7554/eLife.41844.013
**Figure supplement 2.** Ion selectivity and gadolinium block of MmTMEM63A-induced MA currents.
DOI: https://doi.org/10.7554/eLife.41844.014
**Figure supplement 3.** Surface expression of MmTMEM63A.
DOI: https://doi.org/10.7554/eLife.41844.015

humans. PIEZOs are also present in plants and unicellular organisms; however, evidence that they are mechanosensitive in these species is still lacking.

The existence of 15 members of OSCAs in *Arabidopsis thaliana* raises the possibility that there might be redundancy, and will make it a challenge to assign function to these proteins individually. However, *Yuan et al., 2014* reported that under hyperosmotic stress, mutant OSCA1.1 plants had

stunted leaf and root growth (*Yuan et al., 2014*). These phenotypes could indeed be a consequence of impaired mechanotransduction. Expression profile of mouse TMEM63A in public databases (BioGPS.org) suggests expression in tissues that experience mechanical forces such as kidney and stomach, while TMEM63B is high in the nervous system (including dorsal root ganglia) as well as in the heart, skeletal muscle, and stomach. Future studies will investigate the contribution of members of OSCA and TMEM63 families to mechanotransduction in various species.

We note the publication of a recent complementary study (*Zhang et al., 2018*). Our manuscript was previously deposited on BioRxiv (DOI: https://doi.org/10.1101/408732).

# Materials and methods

## Key resources table

| Reagent type (species) or resource | Designation | Source or reference | Identifiers | Additional information |
|---|---|---|---|---|
| Cell line (Piezo1 knockout HEK293T cells) | HEK-P1KO cells | PMID: 28426961 | | Patapoutian Lab (Scripps Research) |
| Antibody | 9E11 anti-Myc antibody (primary) | Santa Cruz Biotechnology | Cat# sc-47694, RRID:AB_627266 | 1:50 |
| Antibody | Alexa Fluor 488 (secondary) | Invitrogen | Cat# A-21121, RRID:AB_141514 | 1:200 |
| Transfected construct (*Arabidopsis thaliana*) | OSCA1.1 | this paper | NM_178966.1, At4g04340 | Human codon optimized, Gene synthesized from Genewiz |
| Transfected construct (*Arabidopsis thaliana*) | OSCA1.2 | this paper | NM_118333.5, At4g22120 | Human codon optimized, Gene synthesized from Genewiz |
| Transfected construct (*Arabidopsis thaliana*) | OSCA1.8 | this paper | NM_102943.6, At1g32090 | Human codon optimized, Gene synthesized from Genewiz |
| Transfected construct (*Arabidopsis thaliana*) | OSCA2.3 | this paper | NM_110975.5, At3g01100 | Human codon optimized, Gene synthesized from Genewiz |
| Transfected construct (*Arabidopsis thaliana*) | OSCA3.1 | this paper | NM_102773.3, At1g30360 | Human codon optimized, Gene synthesized from Genewiz |
| Transfected construct (*Arabidopsis thaliana*) | OSCA4.1 | this paper | NM_119753.3, At4g35870 | Human codon optimized, Gene synthesized from Genewiz |
| Transfected construct (*Drosophila melanogaster*) | DmTMEM63 | this paper | Dmel_CG11210 | Human codon optimized, Gene synthesized from Genewiz |
| Transfected construct (*Mus musculus*) | MmTMEM63A | ORIGENE | NM_144794, Cat No. MR210748 | |
| Transfected construct (*Mus musculus*) | MmTMEM63B | ORIGENE | NM_198167, Cat No. MR221527 | |

*Continued on next page*

*Continued*

| Reagent type (species) or resource | Designation | Source or reference | Identifiers | Additional information |
|---|---|---|---|---|
| Transfected construct (*Mus musculus*) | MmTMEM63C | ORIGENE | NM_172583, Cat No. MR210738 | |
| Transfected construct (*Homo sapiens*) | HsTMEM63A | ORIGENE | NM_014698, Cat No. RC206992 | |

## Generation of clones

The different OSCA clones were gene synthesized (human codon optimized) from Genewiz. Sequences for *Arabidopsis thaliana* OSCA1.1 (NM_178966.1, At4g04340), OSCA1.2 (NM_118333.5, At4g22120), OSCA1.8 (NM_102943.6, At1g32090), OSCA2.3 (NM_110975.5, At3g01100), OSCA 3.1 (NM_102773.3, At1g30360), and OSCA4.1 (NM_119753.3, At4g35870) were downloaded from TAIR (www.arabiopsis.org, RRID:SCR_004618). The synthesized cDNA was cloned into pIRES2-mCherry vector. In addition, OSCA1.1, OSCA1.2, and OSCA3.1 cDNA were cloned from *Arabidopsis thaliana* into pIRES2-mCherry vector. The OSCA1.1, OSCA1.2, and OSCA3.1 codon region were amplified from Arabidopsis cDNA with following primers:

OSCA1.1 Forward primer: ccgctagcgctaccggactcagatcATGGCAACACTTAAAGAC
OSCA1.1 Reverse primer: gggcccgcggtaccgtcgactgcagCTAGACTTCTTTACCGTTAATAAC
OSCA1.2 Forward primer: ccgctagcgctaccggactcagatcATGGCGACACTTCAGGATATTG
OSCA1.2 Reverse primer: gggcccgcggtaccgtcgactgcagTTAGACTAGTTTACCACTAAAGGG
OSCA3.1 Forward primer: gattaacagaagcttcccggCATGGAGTTTGGATCTTTTCTTGTG
OSCA3.1 Reverse primer: gcccttgctcaccatgagctCAACGCCTGCTATTGCGTTG

pIRES2-mCherry plasmid was cut with FastDigest EcoRI and XhoI restriction enzyme (Thermo Fischer Scientific). Then the purified PCR product ligated into the digested plasmid using Gibson assembly kit (NEB). The sequence of genes verified by sequencing. The protein sequence downloaded from TAIR matched the sequence obtained from the plant. Furthermore, MA currents recorded from either clones were indistinguishable in their properties. Therefore, data for OSCA1.1, OSCA1.2, and OSCA3.1 was combined from gene synthesized cDNA and the cDNA sub-cloned from plants. *Drosophila melanogaster* TMEM63 (Dmel_CG11210, GenBank: AAF59136.1) was gene synthesized according to the sequence from GenBank. Mammalian TMEM63 clones were purchased from ORIGENE; MmTMEM63A (NM_144794, Cat No.: MR210748), MmTMEM63B (NM_198167, Cat No.: MR221527), MmTMEM63C (NM_172583, Cat No.: MR210738), and HsTMEM63A (NM_014698, Cat No.: RC206992). Some clones had a Myc tag when purchased, which was removed using QuickChange II XL site-directed mutagenesis kit according to the manufacturer's instruction. The ORF of MmTMEM63A and MmTMEM63B was sub-cloned into pIRES2-mCherry vector, and MmTMEM63C was sub-cloned into pcDNA3.1-IRES-GFP. All clones were full-length sequence verified before testing.

## Cell culture and transient transfection

PIEZO1-knockout Human Embryonic Kidney 293T cells (HEK-P1KO, original HEK293T cell RRID: CVCL_0063)) were used for all heterologous expression experiments. HEK-P1KO cells were generated in house using CRISPR–Cas9 nuclease genome editing technique as described previously (*Dubin et al., 2017*; *Lukacs et al., 2015*), and were negative for mycoplasma contamination. Cells were grown in Dulbecco's Modified Eagle Medium (DMEM) containing 4.5 mg.ml$^{-1}$ glucose, 10% fetal bovine serum, 50 units.ml$^{-1}$ penicillin and 50 µg.ml$^{-1}$ streptomycin. Cells were plated onto 12 mm round glass poly-D-lysine coated coverslips placed in 24-well plates and transfected using lipofectamine 2000 (Invitrogen) according to the manufacturer's instruction. All plasmids were transfected at a concentration of 600 ng.ml$^{-1}$. Cell were recorded from 24 to 48 hr after transfection. Since HsTMEM63A was in a non-fluorescent tagged vector, it was co-transfected with IRES-GFP or pIRES2-mCherry vector.

## Electrophysiology

Patch-clamp experiments in cells and in liposomes were performed in standard whole-cell, cell-attached, or excised patch (outside-out for cells, inside-out for liposomes) mode using Axopatch 200B amplifier (Axon Instruments). Some whole-cell recordings were done using Axon Multiclamp700A. Currents were sampled at 20 kHz and filtered at 2 kHz or 10 kHz. Leak currents before mechanical stimulations were subtracted off-line from the current traces. Voltages were not corrected for a liquid junction potential (LJP) except for ion selectivity experiments. LJP was calculated using Clampex 10.6 software. All experiments were done at room temperature. All electrophysiology data was analyzed in Clampfit 10.6 and data was plotted using GraphPad Prism (RRID:SCR_002798).

## Solutions

For whole-cell patch clamp recordings, recording electrodes had a resistance of 2 – 3 M$\Omega$ when filled with internal solution composed of (in mM) 133 CsCl, 1 CaCl$_2$, 1 MgCl$_2$, 5 EGTA, 10 HEPES (pH 7.3 with CsOH), 4 MgATP and 0.4 Na$_2$GTP. The extracellular solution, also used as ios-osmotic solution, was composed of (in mM) 133 NaCl, 3 KCl, 2.5 CaCl$_2$, 1 MgCl$_2$, 10 HEPES (pH 7.3 with NaOH) and 10 glucose, 300 mmol/kg. Hyperosmolarity solution composed of (mM) 133 NaCl, 3 KCl, 2.5 CaCl$_2$, 1 MgCl$_2$, 10 HEPES (pH 7.3 with NaOH) and 300 Sorbitol, 620 mmol.kg$^{-1}$.

For cell-attached patch clamp recordings, external solution used to zero the membrane potential consisted of (in mM) 140 KCl, 1 MgCl$_2$, 10 glucose and 10 HEPES (pH 7.3 with KOH). Recording pipettes were of 1 – 3 M$\Omega$ resistance when filled with standard solution composed of (in mM) 130 mM NaCl, 5 KCl, 1 CaCl$_2$, 1 MgCl$_2$, 10 TEA-Cl and 10 HEPES (pH 7.3 with NaOH). For gadolinium inhibition experiments, 100 mM GdCl$_3$ stock solution of was diluted in cell-attached pipette solution at a working concentration of 60 $\mu$M.

Ion selectivity experiments for OSCAs were performed in outside-out patch configurations. $P_{Cl}$/$P_{Na}$ was measured in extracellular solution composed of (in mM) 30 NaCl, 10 HEPES and 225 Sucrose (pH 7.3 with NaOH) and intracellular solution consisted of (in mM) 150 NaCl and 10 HEPES (pH 7.3 with NaOH). Ion selectivity experiments (*Figure 4—figure supplement 2*) on MmTMEM63A were performed in cell-attached patch clamp configuration. Independent cells were recorded under different conditions. NMDG-Cl solution consisted of (in mM) 150 NMDG (N-methyl-D-glucamine), 10 HEPES (pH 7.5). KCl solution consisted of (in mM) 150 KCl, 10 HEPES (pH 7.5). Cs-Meth solution consisted of (in mM): 149 Cs-methanesulphonate, 1 CsCl, 10 HEPES (pH 7.5).

## Mechanical stimulation

For whole-cell recordings, mechanical stimulation was achieved using a fire-polished glass pipette (tip diameter 3 – 4 $\mu$m) positioned at an angle of 80° relative to the cell being recorded. Downward displacement of the probe towards the cell was driven by Clampex-controlled piezo-electric crystal microstage (E625 LVPZT Controller/Amplifier; Physik Instrumente). The probe had a velocity of 1 $\mu$m.ms$^{-1}$ during the ramp phase of the command for forward movement and the stimulus was applied for 150 ms. To assess the mechanical sensitivity of a cell, a series of mechanical steps in 0.5 or 1 $\mu$m increments was applied every 10 – 20 s. Threshold was calculated as the differential (y-x) of the probe distance that first touches the cell (x) and the probe distance that induces the first channel response (y).

Macroscopic stretch-activated currents were recorded in the cell-attached or excised, outside-out patch clamp configuration. Membrane patches were stimulated with 500 ms (for OSCA clones) or 1 s or 2 s (for TMEM63 clones) negative or positive pressure pulses through the recording electrode using Clampex controlled pressure clamp HSPC-1 device (ALA-scientific), with inter-sweep duration of 1 min. Since TMEM63 had slower gating kinetics, longer stimulus duration was picked. Negative pressure was applied when patch was in cell-attached and inside-out configuration, positive pressure was applied when patch was in the outside-out configuration. Activation time constant was determined by measuring 10 – 90% rise time (between baseline and peak) for currents at saturating pressure stimulus using Clampfit 10.6.

Stretch-activated single-channel currents were recorded in the cell-attached configuration. Since single-channel amplitude is independent of the pressure intensity, the most optimal pressure stimulation was used to elicit responses that allowed single-channel amplitude measurements. These

stimulation values were largely dependent on the number of channels in a given patch of the recording cell. Single-channel amplitude at a given potential was measured from trace histograms of 5 to 10 repeated recordings. Histograms were fitted with Gaussian equations using Clampfit 10.6 software. Single-channel slope conductance for each individual cell was calculated from linear regression curve fit to single-channel I-V plots. Single-channel current for MscL in proteoliposomes was measured at one membrane voltage (30 mV) and conductance was calculated at that potential assuming 0 pA at 0 mV. In cells, OSCA1.2 single-channel amplitude in 200 mM KCl, 5 mM MOPS, pH 7.0 (KOH) was also measured at −80 mV (again assuming 0 pA at 0 mV) and used to calculate conductance.

## Hyperosmotic stimulation

Hyperosmolarity-induced currents were evoked by two different protocols (*Hou et al., 2014*; *Yuan et al., 2014*). 1) Once in the whole-cell-patch clamp configuration in iso-osmotic solution (300 mmol.kg$^{-1}$) currents were recorded continuously in response to voltage ramps from −100 mV to +100 mV applied every 2 – 10 s. Once a stable response was achieved, the bath solution was switched to hyperosmotic solution (620 mmol.kg$^{-1}$) and currents were recorded for at least five additional minutes. Maximal response at −100 mV in iso-osmotic solution and hyper-osmotic solution were measured for each cell and plotted. 2) In the whole-cell patch clamp configuration whole-cell currents at −80 mV were recorded as cells were first exposed to 1 min of iso-osmotic (300 mmol.kg$^{-1}$) solution followed by 5 mins of hyperosmotic solution (620 mmol.kg$^{-1}$) and back to iso-osmotic solution. Maximal current response in the three conditions were plotted for each cell.

## Permeability ratio measurements

Reversal potential for each cell in the mentioned solution was determined by interpolation of the respective current-voltage data. Permeability ratios were calculated by using the following Goldman-Hodgkin-Katz (GHK) equations:

$P_{Cl}/P_{Na}$ ratios:

$$E_{rev} = \frac{RT}{F} ln \frac{P_{Na}[Na]_o + P_{Cl}[Cl]_i}{P_{Na}[Na]_i + P_{Cl}[Cl]_o}$$

## OSCA1.2- EGFP expression and purification

Codon optimized OSCA1.2 gene were cloned into vector pcDNA3.1. An EGFP tag was placed at the C terminus and connected to the gene via a PreScission Protease cleavable linker (LEVLFQGP) (OSCA1.2-EGFP). A FLAG tag (DYKDDDDK) was added to the C terminus of EGFP with two intervening alanines as a linker. For protein expression and purification, four liters of HEK293F cells (Thermo Fisher Freestyle 293 F, RRID: CVCL_D603) were grown in Freestyle 293 expression media to a density of 1.2 – 1.7 × 106 cells.mL$^{-1}$. All cell lines tested negative for mycoplasma contamination. Each liter was transfected by combining 1 mg.L$^{-1}$ of the construct with 3 mg.L$^{-1}$ of PEI in 30 mL of Opti-MEM and then adding the mix to the culture of cells. Transfected cells were grown for 48 hr and then pelleted, washed with ice cold PBS, flash frozen and stored at −80°C for future use. From this point forward, every step of the purification was carried out at 4°C unless otherwise stated. Pellets were thawed on ice, resuspended in 200 mL of solubilization buffer (25 mM tris pH 8.0, 150 mM NaCl, 1% Lauryl Maltose Neopentyl Glycol (LMNG), 0.1% cholesteryl hemisuccinate (CHS), 2 μg.mL$^{-1}$ leupeptin, 2 μg.mL$^{-1}$ aprotinin, 1 mM PMSF, 2 μM pepstatin, 2 mM DTT) and stirred vigorously for 2 – 3 hr. Subsequently, insoluble material was pelleted via ultracentrifugation for 45 min at 30,000 rpm in a Type 70 Ti rotor. Batch binding of the supernatant was performed for 1 hr with 2 mL of FLAG M2 affinity resin (Sigma) previously washed first with 0.1M glycine pH 3.5 and then with wash buffer (25 mM tris pH = 8.0, 150 mM NaCl, 0.01% LMNG, 0.001% CHS, 2 mM DTT). Resin was placed in a gravity flow column and washed with 10CV of wash buffer. Protein was eluted using 2.5 mL of elution buffer (wash buffer and 200 ug.mL$^{-1}$ 3x FLAG peptide). Presence of protein in the elution was confirmed by SDS-PAGE. Sample was concentrated using a 100 kDa MWCO Amicon Ultra centrifugal filter. Concentrated protein was injected onto Shimadzu HPLC and size exclusion chromatography was performed using a Superose 6 Increase column and wash buffer. Fractions corresponding to OSCA1.2-PP-EGFP were concentrated to 2 mg.mL$^{-1}$.

## EcMscL expression and purification

EcMscL was purchased from Addgene (plasmid # 92418). EcMscL purification was done as previously described (*Rosholm et al., 2017*), with the exception that we performed batch binding with Ni-NTA Agarose (QIAGEN).

## Proteoliposome reconstitution

Protocol for proteoliposome reconstitution was modified from previous publication(*Coste et al., 2012*). Soybean polar lipid extract ((Avanti #541602) was completely desiccated and then resuspended in 200 mM KCl, 5 mM MOPS, pH 7.0 for a final concentration of 10 mg.mL$^{-1}$. The mixture was then bath sonicated for 3 cycles of 2 min sonication followed by 2 min wait. Liposomes were aliquoted and frozen for future use. Liposomes were thawed and supplemented with DDM for a final concentration of 1.5 mM DDM. Protein was diluted to 2 mg.mL$^{-1}$ and was added to liposomes in a 1:100 ratio (protein:lipid). Mixture was placed on ice for 5 min and then incubated at room temperature for 20 min with rotation. 10 mg of previously washed biobeads (one methanol wash, two water washes and one wash with 200 mM KCl, 5 mM MOPS, pH 7.0) were added to the mixture and incubated for 1 hr with rotation at room temperature. Biobeads were removed and a second set of 10 mg of biobeads was added and incubated for 30 min. Biobeads were removed and mixture was centrifuge at 60,000 rpm for 60 min at 8°C in a Beckman Coulter Optima TLX ultracentrifuge.

## Proteoliposome recordings

Protocol used to record stretch-activated currents from proteoliposomes was adapted from *Coste et al., 2012* (*Coste et al., 2012*). The proteoliposome pellet was re-suspended in 40 μl of buffer containing 200 mM KCl, 5 mM MOPS (pH7.0), from which 20 μl drops were placed on a cover slide. The samples were dried under vacuum for >16 hr. Samples were then hydrated with 25 μl of the same buffer and allowed to sit for 2 hr at 4°C before recordings. 2–3 μl of proteoliposomes were withdrawn from the edge of the spots on the cover slide and transferred to the recording chamber. After 5 min, the chamber was slowly filled with recording solution. Multi-GΩ seals were made to proteoliposomes immobilized at the bottom of the recording chamber. At that time, the proteoliposome patch was excised to create an inside-out patch. Pipette and bath solution contained (in mM) 200 KCl, 5 MOPS titrated to pH 7.0 with KOH.

## Surface immunostaining

Surface immunostaining of MmTMEM63A was carried out as described previously (*Coste et al., 2015*), with slight modifications. Briefly, pIRES2-mCherry vector encoding MmTMEM63A construct containing myc-tag (EQKLISEEDL) inserted after the first amino acid were expressed in HEK-P1KO cells on poly-D-lysine treated glass coverslips. 2 days after transfection, labeling of non-permeabilized cells was carried out by incubating the cells with 9E11 anti-Myc antibody (1:50; Santa Cruz Biotechnology, Cat# sc-47694, RRID:AB_627266). After six washes with warm medium, cells were incubated with secondary antibodies conjugated to Alexa Fluor 488 (1:200; Invitrogen Cat# A-21121, RRID:AB_141514) for 10 min at room temperature. Cells were washed six times with warm media and once with PBS, and then fixed with 4% PFA/PBS for 30 min. For permeabilization, cells were fixed with 4% PFA for 10 min and then treated with 0.3% Triton X-100 and blocked with 10% normal goat serum in PBS prior to incubation with antibodies (primary: 1:200 for 2 hr, secondary: 1:400 for 1 hr, in block solution). Cells were imaged with a Nikon C2 confocal microscope with 40 × oil immersion objective. The live labeling and permeabilized staining were repeated in three separate experiments to confirm results.

## Acknowledgment

We thank Drs. Kei Saotome, Jorg Grandl, Viktor Lukacs, Jose Santos, Michael Bandell, and David Ginty, and members of the Patapoutian lab for helpful discussions. We acknowledge Allain Fransisco, Meaghan Loud, Adam Coombs, and Jayanti Mathur for technical support.

# Additional information

## Funding

| Funder | Grant reference number | Author |
|---|---|---|
| National Institute of Neurological Disorders and Stroke | R35NS105067 | Ardem Patapoutian |
| Howard Hughes Medical Institute | | Ardem Patapoutian |
| National Institutes of Health | R21DE025329 | Adrienne E Dubin |
| Ray Thomas Edwards Foundation | | Andrew B Ward |

The funders had no role in study design, data collection and interpretation, or the decision to submit the work for publication.

## Author contributions

Swetha E Murthy, Conceptualization, Data curation, Formal analysis, Methodology, Writing—original draft, Writing—review and editing; Adrienne E Dubin, Data curation, Funding acquisition, Methodology, Writing—review and editing; Tess Whitwam, Sebastian Jojoa-Cruz, Stuart M Cahalan, Seyed Ali Reza Mousavi, Methodology, Writing—review and editing; Andrew B Ward, Supervision, Funding acquisition, Writing—review and editing; Ardem Patapoutian, Conceptualization, Supervision, Funding acquisition, Writing—review and editing

## Author ORCIDs

Swetha E Murthy (ID) http://orcid.org/0000-0001-9580-3380
Adrienne E Dubin (ID) http://orcid.org/0000-0003-4683-7175
Sebastian Jojoa-Cruz (ID) https://orcid.org/0000-0002-4392-3898
Andrew B Ward (ID) http://orcid.org/0000-0001-7153-3769
Ardem Patapoutian (ID) http://orcid.org/0000-0003-0726-7034

## Decision letter and Author response

Decision letter https://doi.org/10.7554/eLife.41844.018
Author response https://doi.org/10.7554/eLife.41844.019

# Additional files

## Supplementary files

• Transparent reporting form
DOI: https://doi.org/10.7554/eLife.41844.016

## Data availability

All data generated or analyzed during this study are included in the manuscript.

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
