## [Decision Letter]

Congratulations, we are pleased to inform you that your article, "OSCA/TMEM63 are an Evolutionarily Conserved Family of Mechanically Activated Ion Channels", has been accepted for publication in *eLife*.

This important manuscript describes the mechano-sensitivity of OSCA channels in plants, *Drosophila* and humans. This careful and elegant study reveals that OSCA is a stretch-activated high threshold non-selective cationic channel. Structure of the plant OSCA channels recently appeared in Nature Structural and Molecular Biology (as stated by the authors). However, this study only provided limited information about the electrophysiological characterization of this new class of force-gated ion channels. The study of Patapoutian and colleagues provides a more in-depth analysis of the biophysical properties of OSCA channels and most importantly demonstrate the mechanosensitivity of the human isoforms. Notably, human OSCA channels are non-inactivating, unlike PIEZO1/2 channels (the other class of MS cationic channels). Another key point is the demonstration that OSCA is activated by stretch when reconstituted into artificial bilayer, demonstrating that the mechanosensitivity is intrinsic to OSCA.

Below is a list of minor comments:

1) Table 1. Why not add the channel conductance for MmPIEZO1 in the last column of the Table. From Coste et al. (2015), it was 29 pS under normal ionic conditions.

2) It would be useful to provide the ionic selectivity of OSCA1. Yuan et al. imply the OSCA1 are K^+^ channels with some permeability to Ca^2+^ which seems curious. This information is pertinent to function.

3) Figure 1—figure supplement 3 and Results, third paragraph. Give a reason for thinking these are sub-conductance states rather than two separate channels in the cell-attached patch. This is important because the conclusion is used to argue the channels are dimeric. Can amplitude histograms be provided?

4) In stimulating with a glass probe, the notion of threshold is used, OSCA1 having a higher displacement threshold than PIEZO1 (Results, second paragraph). I do not understand what threshold signifies. Is this measured from the point at which the probe touches the membrane, and does it reflect the extent to which the cell body is indented before channels are opened? Perhaps comment.

5) The authors use the term "isoform" to describe products of different genes. The term "isoform" is frequently used to describe alternative spliced variants of a gene. To avoid confusion, it might be useful to change this to something like "homologs encoded by different genes" or "family members encoded by different genes".

6) A recent paper describes the structure of OSCA channels, indicating that they are dimers. This could explain the subconductance states observed by the authors. They might want to mention this in the manuscript.

7) The currents carried by the *Drosophila* and mammalian orthologs seem small compared to OSCAs (Imax 14-19 pA versus 100s-1000s of pA depending on stimulus conditions). Did the proteins traffic effectively to the cell surface of the cell line? Is the small current reflective of the true properties of the *Drosophila*/mammalian proteins or could the cell lines lack accessory proteins that are required for channel gating? The authors might want to discuss this point more thoroughly.

---

## [Author Response]

Below is a list of minor comments:1) Table 1. Why not add the channel conductance for MmPIEZO1 in the last column of the Table. From Coste et al. (2015,) it was 29 pS under normal ionic conditions.

We have now added values for MmPIEZO1 single-channel conductance in Table 1.

2) It would be useful to provide the ionic selectivity of OSCA1. Yuan et al. imply the OSCA1 are K^+^ channels with some permeability to Ca^2+^ which seems curious. This information is pertinent to function.

We report in the paper that OSCA1.1 and OSCA1.2 are cation selective, with 21 to 17% permeability to chloride. However, to evaluate selectivity to different cations in OSCA and TMEM63 proteins, a detailed analysis of selectivity experiments for different monovalent and divalent cations is required. Though we agree that this is important for function, we think it is beyond the scope of this study.

3) Figure 1—figure supplement 3 and Results, third paragraph. Give a reason for thinking these are sub-conductance states rather than two separate channels in the cell-attached patch. This is important because the conclusion is used to argue the channels are dimeric. Can amplitude histograms be provided?

Amplitude histograms are now added to Figure1—figure supplement 2. The frequency of the intermediate state was always low (10%) compared to the full open state, which was why we speculated a dimeric channel. Consistently, the cryo-EM structure of OSCA1.1 (Zhang et al., 2018) and OSCA1.2 (Jojoa-Cruz et al., 2018) reveal a pore within each subunit of a dimeric channel.

4) In stimulating with a glass probe, the notion of threshold is used, OSCA1 having a higher displacement threshold than PIEZO1 (Results, second paragraph). I do not understand what threshold signifies. Is this measured from the point at which the probe touches the membrane, and does it reflect the extent to which the cell body is indented before channels are opened? Perhaps comment.

Threshold is calculated as the differential (y-x) of the probe distance that first touches the cell (x) and the probe distance that induces the first channel response (y). Therefore, it is the minimum distance of indentation required to activate the channel. This was mentioned in the Materials and methods section. We now describe this in the main text.

5) The authors use the term "isoform" to describe products of different genes. The term "isoform" is frequently used to describe alternative spliced variants of a gene. To avoid confusion, it might be useful to change this to something like "homologs encoded by different genes" or "family members encoded by different genes".

The text has been edited to replace “isoform” with the appropriate nomenclature.

6) A recent paper describes the structure of OSCA channels, indicating that they are dimers. This could explain the subconductance states observed by the authors. They might want to mention this in the manuscript.

We do mention in the text that the subconductance state is indicative of a dimeric channel, as seen in the OSCA1.2 cryo-EM structure reported in the accompanying paper from our group, as well as OSCA1.1 structure determined by Zhang et al. (2018).

7) The currents carried by the Drosophila and mammalian orthologs seem small compared to OSCAs (Imax 14-19 pA versus 100s-1000s of pA depending on stimulus conditions). Did the proteins traffic effectively to the cell surface of the cell line? Is the small current reflective of the true properties of the Drosophila/mammalian proteins or could the cell lines lack accessory proteins that are required for channel gating? The authors might want to discuss this point more thoroughly.

The reviewer raises a valid concern. We have now included data (Figure 4—figure supplement 4) which indicates that mouse TMEM63A is expressed on the membrane, suggesting that the small stretch-activated currents may not be due to trafficking issues. In addition, stretch-activated current properties of fly and mammalian TMEM63 are strikingly similar to plant OSCA2.3 (small macroscopic peak currents, unresolvable single-channel currents, and slow inactivation kinetics), though they share only ~20% identity. Therefore, it is highly unlikely that the lack of accessory proteins in HEK-P1KO cells is the source of disparate gating properties. However, one cannot exclude the possibility that in its native cellular environment, other auxiliary subunits might interact with OSCA2.3 and TMEM63 proteins to alter the channel’s permeation and gating properties. Future studies will test this hypothesis. This discussion is now included in the manuscript.